# Impact and cost-effectiveness of the community-led AMETHIST intervention among female sex workers in Zimbabwe

Loveleen Bansi-Matharu [1] ✉, Collin Mangenah[2,3], Paul Revill[4], Fortunate Machingura [2,3], Sanni Ali[5], Sungai T. Chabata [2,3], Primrose Matambanadzo[2,3], Richard Steen[6], Amon Mpofu[7], Owen Mugurungi[8], Isaac Taramusi[9], James R. Hargreaves[5], Frances M. Cowan [2,3] & Andrew N. Phillips [1]

Female sex workers (FSW) face high HIV risk of HIV transmission and acquisition. The AMETHIST ("Adapted Microplanning: Eliminating Transmissible HIV In Sex Transactions") trial enhanced Zimbabwe's Key Populations (KP) programme by providing targeted, community-based support for FSW. We used the HIV Synthesis Model to assess its long-term impact and cost-effectiveness. Given USAID's major role in funding, we also evaluated the effects of ending US support on the KP programme. We modelled a KP programme from 2010. From 2024 we compared (i) continuation of KP programme to (ii) continuation of KP programme + 'AMETHIST' intervention. We assessed HIV outcomes in 2030 and conducted cost-effectiveness analysis over a 50 year time horizon. Similar analyses were undertaken comparing continuation of the current KP programme to discontinuation. Here we show that AMETHIST had greater positive impact than the KP programme alone; a higher proportion of FSW tested for HIV in the past year, were diagnosed, on ART and had undetectable viral loads compared to the KP programme alone. Disability adjusted life years were averted with AMETHIST and it was cost-saving within 15 years. Continuing the current KP programme was also cost-saving compared to discontinuation of the KP programme.

HIV incidence at a population level is declining in East, Central and South Africa (ECSA) but incidence amongst female sex workers (FSW) is still up to 9 times higher than females in the general population[1]. FSW usually account for less than 5% of the wider population[2] and those living with HIV are at high risk of transmitting HIV to their sexual partners if not adequately engaged with dedicated support services[2,3]. FSW who are HIV negative are at high risk of HIV acquisition if pre-exposure prophylaxis (PrEP) or condom use is sub-optimal.

Though HIV programs specifically for FSW are limited in the region, where they do exist, they can make considerable impact on prevention services and on risk of onward transmission through linkage to care and adherence support[4,5]. One such programme is the Key Populations (KP) (formerly Sisters) programme, Zimbabwe's nationally

---

[1]Institute for Global Health, University College London, London, UK. [2]Centre for Sexual Health and HIV AIDS Research (CeSHHAR) Zimbabwe, Harare, Zimbabwe. [3]International Public Health Department, Liverpool School of Tropical Medicine, Liverpool, UK. [4]Centre for Health Economics, University of York, York, UK. [5]Department of Population Health, London School of Hygiene and Tropical Medicine, London, UK. [6]Department of Public Health, Erasmus University, Rotterdam, Netherlands. [7]National AIDS Council, Harare, Zimbabwe. [8]AIDS and TB Directorate, Ministry of Health and Child Care, Harare, Zimbabwe. [9]UNAIDS UCO Zimbabwe, Harare, Zimbabwe. ✉e-mail: l.bansi-matharu@ucl.ac.uk

**Table 1 | Baseline comparison of modelled and observed data of FSW**

| | | Modelled outputs in 2024 Mean (90% range) | Observed data (Usual care AMETHIST (2022) arm[7] unless otherwise stated) Mean (95% CI) |
|---|---|---|---|
| Number of SW | | 63420 (15569, 118412) | 70423 (59271–79518)[20] |
| Percentage of women who are currently SW | | 1.6 (0.4, 2.9) | 1.6 (1.3–1.8)[20] |
| Percentage of women who have ever been SW 15-49 | | 2.8 (1.3, 4.8) | - |
| Percentage of SW in each age group | 15-19 | 5.7 (2.6, 10.1) | 4.2 (3.3–5.2) (aged 18–19) |
| | 20-24 | 18.4 (12.2, 25.3) | 17.4 (15.8–19.2) |
| | 25-29 | 19.3 (15.0, 26.6) | 19.8 (18.2 – 21.6) |
| | 30-39 | 32.3 (24.6, 39.2) | 36.1 (34.1–38.2) |
| | 40+ | 24.3 (18.2, 31.2) | 22.5 (20.8–24.3) |
| Percentage of SW with age debut in each category | 15-19 | 33.3 (26.0, 40.2) | 33.8 (31.8–35.9) |
| | 20-29 | 48.8 (41.0, 56.2) | 48.1 (46.0–50.3) |
| | 30-39 | 13.0 (8.7, 18.1) | 18.1 (16.5–19.8) (aged 30 plus) |
| | 40+ | 5.0 (2.3, 8.9) | - |
| Percentage of SW with total years spent as SW (years) | 0-2 | 24.5 (17.7, 38.3) | 29.3 |
| | 3-5 | 18.9 (12.7, 24.6) | 20.8 |
| | 6-9 | 18.6 (13.2, 23.6) | 27.7 |
| | 10-19 | 24.6 (13.2, 32.3) | 12.1 |
| Percentage of SW with 0 condomless partners (Includes periods of inactivity) | | 34.9 (10.8, 59.6) | 27.6 (25.7–29.5) |
| Percentage of SW on PrEP | | 10.5 (2.9, 23.2) | 15.7 (min 8.4–max 30.2) |
| HIV Incidence in SW | | 10.02 (0.90, 30.33) | - |
| HIV Prevalence in SW | | 56.9 (34.8, 80.5) | 48.4 (46.2–50.5) (RDS survey) |
| Percentage of SW diagnosed | | 90.9 (82.5, 97.5) | 88.7 (86.7–90.7) |
| Of those diagnosed, percentage of SW on ART | | 96.3 (92.0, 99.2) | 94.4 (92.8–95.7) |
| Of those on ART, percentage of SW with viral suppression | | 91.6 (77.7, 100.0) | 92.8 (90.6–94.3) |
| Percentage of SW who have visited a SW programme | | 57.1 (35.8, 79.5) | 52.6 (50.4–54.8) |

scaled programme for sex workers run on behalf of the Ministry of Health and Child Care and National AIDS Council since 2009[6]. It is estimated that almost 60% of all FSW in Zimbabwe have attended the KP programme[6] which offers enhanced community–based services including HIV testing and counselling, sexual and reproductive health services, condom and PrEP provision, and health education supported by trained peer educators. Of those living with HIV, the proportion of FSW who know their HIV status, are taking antiretroviral therapy (ART) and have suppressed viral loads is similar to that seen in the general population[7]

Targeted HIV related services for sex workers, supported by peer-based community outreach are recommended by UNAIDS/WHO. The "AMETHIST" ("Adapted Microplanning: Eliminating Transmissible HIV In Sex Transactions") trial[7] built on the existing KP programme to deliver intensive and systematic community-based support for sex workers targeted according to their level of vulnerability (peer-led microplanning). The intervention also included self-help groups to build community cohesion amongst members and the wider sex work community. The proportion of FSW at risk of transmitting (HIV-positive, not virally suppressed or not consistently using condoms) or acquiring HIV (HIV-negative and not protected by condoms or PrEP) was compared amongst those in the intervention group and those in the usual care group. Whilst there was no impact on the primary outcome due to there being no effect on the proportion of HIV negative sex workers at risk of acquiring HIV, the key secondary outcome of the proportion of FSW living with HIV at risk of transmitting HIV was significantly reduced in the intervention arm (−5·5% (95% CI −8·2% to −2·9%))

The KP programme in Zimbabwe has been operating cost-effectively at a cost of $US 8.5 million per year[8], and would still be considered cost-effective if the overall cost was no higher than $34 million per year/10 million adults[8]. Using data from the AMETHIST

trial, including the cost of the intervention and the impact it had on those living with HIV, we used the HIV Synthesis Model to assess the longer term impact of AMETHIST on key HIV associated outcomes and conducted a cost-effectiveness analysis to help inform policy decisions specific to FSW.

These analyses were planned and conducted prior to the USAID Stop-Work Order, which mandates the immediate suspension of all USAID-funded activities. Funding for the KP programme has since been significantly reduced and in light of the uncertainty ahead, we also looked at the impact of the existing KP programme being fully defunded, i.e., we compared key HIV outcomes and conducted cost-effectiveness analyses comparing the existing KP programme to a counterfactual scenario of there being no dedicated programme for FSW in Zimbabwe.

## Results

The modelled number and characteristics of FSW are shown in Table 1, along with observed data from the AMETHIST study. Modelled estimates align well with empirical data, both in terms of the national estimate of the number of FSW and more granular outputs such as the age at start of sex work, duration of sex work, percentage of sex workers who have visited a sex worker programme and HIV prevalence amongst FSW (mean 56.9% (90% range: 34.8, 80.5) modelled, 48.4 empirical). The 90% ranges for the modelled proportion of FSW diagnosed with HIV, on ART and virally suppressed were in line with point estimates observed in the usual care of the AMETHIST trial (modelled mean estimates: 90.9% (82.5, 97.5), 96.3% (92.0, 99.2), 91.6% (77.7, 100.0) respectively, empirical estimates: 88.7%, 94.4%, 92.8% respectively).

When modelling continuation of the Sister's programme, we found in 2030, 58.0% (34.4, 80.4) FSW had visited a sex worker

programme (Table 2). 61.9% (46.0, 82.7) of all HIV negative FSW had had a HIV test in the past year and the proportion of FSW diagnosed, on treatment and virally suppressed was 94.1% (87.0, 99.3), 96.1% (89.4, 100.0) and 92.5% (78.2, 99.8) respectively. 34.8% (11.3, 58.3) of FSW were estimated to have 0 condomless partners in any 3-month period (this included those who may currently not be in active sex work). PrEP uptake was 25.1% (9.5, 42.0) and HIV incidence was estimated to be 5.80 (0.01, 17.78)/100 person years (py). Incidence in the general population was estimated to be 0.25 (0.08, 0.56) /100 py. The AMETHIST intervention modelled from 2024 onwards improved all outcomes, albeit to different magnitudes (Table 2). In 2030, a slightly higher proportion of sex workers visited a sex worker programme (60.1% (38.6, 81.0)) and 74.2% (59.1, 88.9) of negative FSW had been tested for HIV in the past year. The proportion of FSW diagnosed, on ART and virally suppressed increased to 95.8% (89.6, 100.0), 96.9% (92.9, 100.0) and 93.3% (82.8, 100.0) respectively. There was a marginal difference in the proportion of FSW with 0 condomless partners (35.2% (11.6, 59.9)) and 30.4% (15.7, 50.2) of HIV negative FSW were estimated to be on PrEP. Incidence in FSW was estimated to be lower in 2030 with the AMETHIST intervention; 5.59 (0, 15.90)/100 py, and there was also a small decline in the incidence amongst the general population (mean difference: −0.02 [−0.17, 0.13] /100 py).

Taking into account the additional cost of AMETHIST per year per FSW seen, overall costs across a 50-year time horizon were lower with the AMETHIST intervention compared to the KP programme ($US −1.9 (−2.6, −1.1) million) (Table 3). A full breakdown of these costs can be found in the Appendix, Table A2. DALYs were also lower with the AMETHIST intervention (9973 (6882, 13064) DALYs averted), resulting in AMETHIST being a cost-saving intervention (and is therefore cost-effective using the $500/DALY cost effectiveness threshold). Moreover, the intervention resulted in lower costs and averted DALYs within 15 years (by 2040) of it being introduced (Fig. 1a).

When comparing continuation of the existing KP programme to discontinuation of the programme, we found key HIV outcomes for FSW were considerably worse in 2030 for FSW when the programme was discontinued (Table 4). The proportions of FSW diagnosed with HIV, on ART and virally suppressed were 91.4% (84.0, 97.1), 85.3% (64.0, 99.0) and 49.9% (24.3, 97.0) respectively. Incidence in FSW rose from 5.80 (0.01, 17.78)/100 py with the KP programme in place to 9.26 (0.29, 28.50)/100 py if the programme was discontinued; consequently incidence in the general population rose to 0.37 (0.11, 0.77)/ py with discontinuation of the KP programme (from 0.25 (0.08, 0.56)/100py). Over a 50 year time horizon, the existing KP programme averted DALYs and had lower costs compared to discontinuation of the programme resulting in the existing KP programme being cost-saving. This cost-saving was seen within 15 years (by 2040) of the 50 year time horizon considered.

## Discussion

Using the HIV Synthesis Model, we were able to calibrate well to baseline empirical data on female sex workers in Zimbabwe in the context of the ongoing implementation of the KP programme. We further modelled the incremental impact of the AMETHIST intervention in line with study results[7]; informed by the trial, we assumed no impact of the intervention on primary prevention of HIV in FSW but did capture the observed impact on female sex workers living with HIV over and above the existing programme. Using parameter values informed by the AMETHIST trial results we projected that if the AMETHIST intervention were to be rolled out in 2024, then by 2030 FSW were more likely to be tested for HIV, diagnosed, receiving ART and virally suppressed compared to the KP programme alone. We did not calibrate to HIV incidence amongst FSW. However, our modelled outcomes in 2030 suggested that HIV incidence decreased with the AMETHIST intervention compared to the KP programme alone. Our

cost effectiveness analyses showed the AMETHIST intervention was cost saving when considered across a 50-year time period. Conversely, if the existing KP programme were to be discontinued, key HIV outcomes for FSW were considerably worse compared to continuation of the programme. Further, continuation of the KP programme was cost-saving (when considering costs and DALYs at a population level – not just restricted to FSW) compared to discontinuation of the current programme.

In the trial, the effect of the AMETHIST intervention on the HIV prevention component of the composite endpoint was non-significant. This was mimicked in the modelling; there was no additional impact of PrEP or condom use considered in the AMETHIST scenarios. However, by reaching even a small number more of FSW, it is likely that the overall proportion of SW currently on PrEP will increase indirectly with the AMETHIST intervention as seen in our modelled results (25% KP vs. 30% AMETHIST) [and also in the trial itself in relation to self-reported uptake of PrEP].

Our results are likely to be conservative. For example, we assumed no impact on condom use with the AMETHIST intervention in line with findings from the trial. The trial took place during the COVID-19 pandemic which inevitably impacted on service provision. We expect the number of additional SW seen under AMETHIST was considerably lower than would have been otherwise; hence the modelled outcomes for SW likely represent the lower end of impact. Further, the trial had stringent criteria in relation to defining condom use including classifying a woman who reports any condomless sex with a long term partner as having inconsistent condom use. As stated in the Discussion of the trial results, this could be considered too stringent when considering the risk of HIV acquisition. Whilst consistent condom use can be challenging in sex work, even a small increase in condom use, particularly amongst high risk FSW, is likely to positively impact outcomes for FSW[9–11] (which may be possible through tailored condom provision offered through microplanning and condom health education) which from a cost effectiveness perspective is likely to further increase the cost-savings.

Our modelling is limited by the nature of the data we are calibrating to. In particular, data on sexual behaviour and condom use is based on self-reports which are prone to bias. This is evidenced in the AMETHIST trial itself where data on self-reported current PrEP use was at odds with plasma concentrations of tenofovir diphosphate and self reported consistent condoms use was at odds with presence of Y chromosome[7]. Hence there is considerable uncertainty surrounding reports of condom use. We attempt to reflect this uncertainty within our modelling by sampling key parameters and report averages with uncertainty ranges. Of note, our 90% uncertainty range of the proportion of FSW with zero condomless partners ranged from 11% to 60%. Using this approach, we feel we have been able to reasonably model the KP programme evidenced by Table 1 and hence project forwards with the AMETHIST intervention. The trial itself also had notable limitations, including the exclusion of baseline data from the analyses and the reliance solely on endline group comparisons for outcome assessment[7].

Using a risk-differentiated community led intervention to increase engagement and uptake of key services for FSW living with HIV is cost-saving within 15 years of introduction compared to the existing KP programme. The impact we have shown is likely to be conservative; an intervention such as AMETHIST is likely to result in a higher proportion of SW being seen than we have modelled and the introduction of long-acting PrEP is likely to further lower HIV incidence if women can be adequately supported to return for follow-up injections. Further, risk differentiated support may help FSW in areas beyond that we have been able to model such as general health and wellbeing including mental health counselling. Upgrading the existing KP programme in Zimbabwe to a risk-differentiated community led intervention will

**Table 2 | Modelled outputs of 'KP' vs. 'AMETHIST' in 2030**

| | Modelled KP | Modelled AMETHIST | Modelled discontinuation of KP | Mean difference between KP and AMETHIST | Observed difference between KP and AMETHIST in 2022 |
|---|---|---|---|---|---|
| | Mean (90% range) | Mean (90% range) | Mean (90% range) | (95% CI) [90% range] | Mean |
| Proportion of all SW visiting programme | 58.0 (34.4, 80.4) | 60.1 (38.6, 81.0) | 0 | 2.1 (1.0, 3.2) [-6.3, 12.3] | |
| Proportion of HIV negative SW tested in past year | 61.9 (46.0, 82.7) | 74.2 (59.1, 88.9) | 60.2 (42.9, 83.1) | 12.3 (10.5, 14.2) [-3.9, 26.1] | |
| Proportion of SW living with HIV diagnosed | 94.1 (87.0, 99.3) | 95.8 (89.6, 100.0) | 91.4 (84.0, 97.1) | 1.7 (0.9, 2.5) [-4.5, 7.4] | 1.0% |
| Of those diagnosed, proportion of SW on ART | 96.1 (89.4, 100.0) | 96.9 (92.9, 100.0) | 85.3 (64.0, 99.0) | 0.8 (0.2, 1.3) [-2.6, 5.9] | 1.8% |
| Of those on ART, proportion of SW virally suppressed | 92.5 (78.2, 99.8) | 93.3 (82.8, 100.0) | 49.9 (24.3, 97.0) | 0.7 (-0.2, 1.7) [-6.1, 8.3] | 4.0% |
| Proportion of all SW with 0 condomless partners | 34.8 (11.3, 58.3) | 35.2 (11.6, 59.9) | 30.2 (8.9, 46.9) | 0.4 (-0.5, 1.3) [-5.8, 6.4] | - |
| Proportion of HIV negative SW on PrEP* | 25.1 (9.5, 42.0) | 30.4 (15.7, 50.2) | 21.8 (9.3, 35.3) | 5.3 (4.0, 6.6) [-8.7, 13.9] | 6.7% |
| Incidence in SW | 5.80 (0.01, 17.78) | 5.59 (0, 15.90) | 9.26 (0.29, 28.50) | -0.21 (-0.93, 0.51) [-7.09, 5.39] | - |
| Prevalence in SW | 46.2 (20.8, 76.8) | 46.2 (23.1, 71.2) | 49.4 (23.7, 76.5) | 0.0 (-1.0, 1.1) [-10.0, 6.7] | -5.7% |
| **General population** | | | | | |
| Incidence | 0.25 (0.08, 0.56) | 0.24 (0.08, 0.47) | 0.37 (0.11, 0.77) | -0.02 (-0.03, 0.00) [-0.17, 0.13] | |
| Prevalence | 7.3 (5.1, 9.8) | 7.2 (4.6, 9.7) | 7.7 (5.3, 10.3) | -0.07 (-0.12, -0.01) [-0.5, 0.4] | |
| Proportion diagnosed | 93.1 (89.3, 96.7) | 93.5 (89.5, 97.1) | 91.4 (86.3, 95.7) | 0.2 (-0.0, 0.4) [-1.4, 1.7] | |
| Of those diagnosed, proportion on ART | 97.1 (94.7, 98.7) | 97.5 (94.7, 98.6) | 96.7 (94.1, 98.5) | 0.1 (0.0, 0.2) [-0.4, 0.6] | |
| Of those on ART, proportion virally suppressed | 93.8 (86.4, 98.3) | 93.8 (87.3, 98.4) | 92.9 (84.9, 97.9) | -0.0 (-0.1, 0.1) [-0.9, 0.8] | |

*Injectable PrEP is assumed to be available from 2027.

**Table 3 | Cost effectiveness outputs comparing KP and KP + AMETHIST (All costs/Disability adjusted life years (DALYs) are discounted at 3% per year, over a 50 year time horizon in the context of 10 million adults)**

| Costs/DALYs are given per year, over a 50 year time Horizon | Continuation of KP from 2024 onwards Mean across runs (95% CI) | KP + AMETHIST from 2024 onwards Mean across runs (95% CI) |
|---|---|---|
| Overall costs in US$ millions assuming $132/year for KP and $155/year AMETHIST (based on costs collected in AMETHIST) | 156 (153, 160) | 154 (151, 158) |
| **Difference in overall costs** | | **−1.9 (−2.6, −1.1)** |
| ART costs | 40 (38, 42) | 38 (37, 40) |
| Difference in ART costs | | −1.3 (−1.6, −1.1) |
| Testing costs | 17 (16, 18) | 17 (16, 19) |
| Difference in testing costs | | 0.1 (0.0, 0.3) |
| Other programme costs | 99 (97, 102) | 99 (96, 101) |
| Difference in other programme costs | | −0.7 (−1.1, −0.2) |
| Testing costs amongst SW | 0.02 (0.02, 0.02) | 0.14 (0.12, 0.17) |
| Difference in testing costs amongst SW | | 0.12 (0.11, 0.14) |
| DALYs | 1786127 (1768200, 1804055) | 1776155 (1758323, 1793986) |
| **Difference in DALYs compared to KP** | | **−9973 (−13064, −6882)** |
| Net DALYs (DALYs + Costs/$500) | 2099035 (2076774, 2121295) | 2085317 (2062869, 2107766) |
| **Difference in Net DALYs** | | **−13717 (−17316, −10119)** |

Bold: key results.

a)

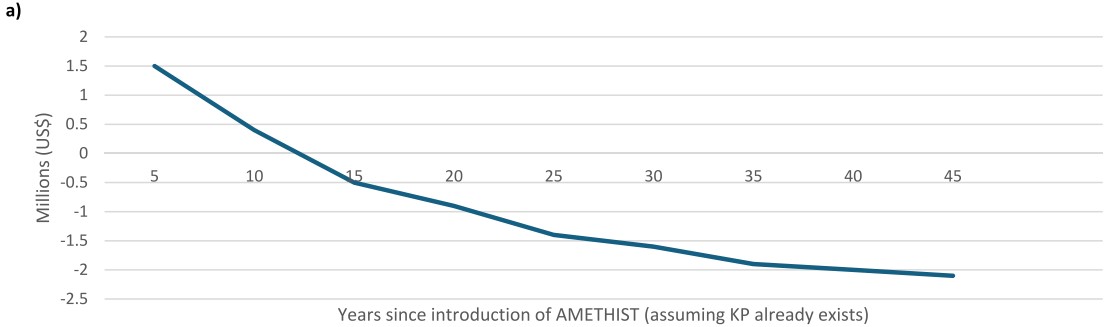

Years since introduction of AMETHIST (assuming KP already exists)

b)

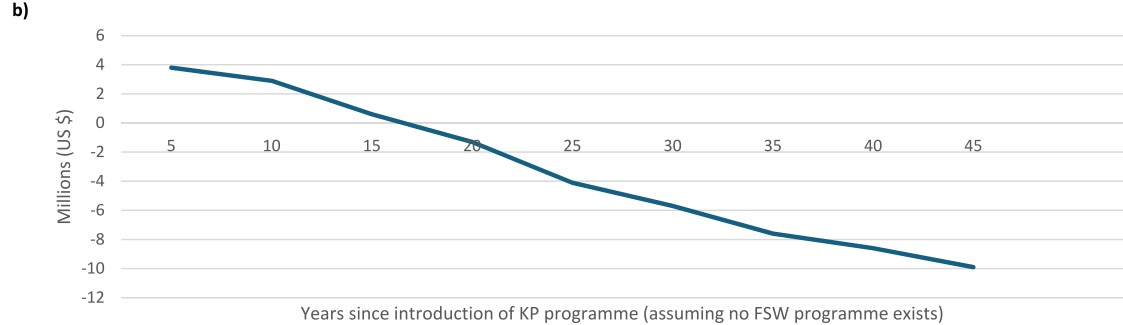

Years since introduction of KP programme (assuming no FSW programme exists)

**Fig. 1 | Additional costs of HIV support programmes. a** Additional costs of AMETHIST assuming KP programme already exists; (**b**) Additional costs of KP programme assuming no FSW programme exists.

reduce HIV incidence in FSW (via a higher proportion of FSW on PrEP and effective treatment) and by increasing viral suppression rates amongst this key population, the risk of onward transmission can be considerably reduced. This ultimately results in lower incidence in the general population. Risk differentiated community led interventions for FSW in addition to existing sex worker programs are also likely to be cost-effective in countries with existing SW programs in the region[8]. Conversely, defunding the existing KP programme will result in higher

costs and an increase in DALYs; this should be taken into account when determining funding allocations.

## Methods

The HIV Synthesis Model, which has been previously used to assess the impact of sex worker programmes on key HIV outcomes in ECSA[8], is an individual-based stochastic simulation model of HIV transmission, progression and the effect of ART. Each time the model is run, a

**Table 4 | Cost effectiveness outputs comparing continuation of the current KP programme and discontinuation of the KP programme (All costs/Disability adjusted life years (DALYs) are discounted at 3% per year, over a 50 year time horizon in the context of Zimbabwe)**

| Costs/DALYs are given per year, over a 50 year time horizon | Continuation of KP from 2024 onwards Mean across runs (95% CI) | KP + AMETHIST from 2024 onwards Mean across runs (95% CI) |
|---|---|---|
| Overall costs in US$ millions assuming $132/year for KP and $155/year AMETHIST (based on costs collected in AMETHIST) | 156 (153, 160) | 165 (160, 169) |
| **Difference in overall costs** | | **−8.1 (−10.0, −6.3)** |
| ART costs | 40 (38, 42) | 45 (43, 47) |
| Difference in ART costs | | −4.8 (−5.6, −3.9) |
| Testing costs | 17 (16, 18) | 18 (16, 19) |
| Difference in testing costs | | −0.5 (−0.8, −0.2) |
| Cost of visiting a SW program | 4 (4, 4) | 0 |
| Different in cost of visiting a SW program | | 4 (4, 4) |
| Testing costs amongst SW | 0.02 (0.02, 0.02) | 0 |
| Difference in testing costs amongst SW | | 0.02 (0.02, 0.02) |
| DALYs | 1786127 (1768200, 1804055) | 1802332 (1782871, 1821792) |
| **Difference in DALYs compared to KP** | | **−16204 (−20912, −11497)** |
| Net DALYs (DALYs + Costs/$500) | 2099035 (2076774, 2121295) | 2131488 (2106673, 2156301) |
| **Difference in Net DALYs** | | **−32453 (−39899, −25007)** |

Bold: key results.

simulated population of adults is generated. Variables are updated every 3 months from 1989 and include age, sex, the number of short term condomless sex partners, presence of a long term condomless sex partner, use of pre-exposure prophylaxis (PrEP), HIV testing, HIV acquisition and additionally, in people living with HIV (PLHIV), viral load, CD4 count, use of specific ART drugs, adherence to ART, resistance to specific drugs and risk of HIV-related death. A 3-month time period has been used in this model to achieve a balance between capturing key updates of values of variables for each individual and model running computing time. SAS 9.4 was used to code the model. Full details of the model can be found in the Appendix, page 5, supported by a short video and slideset ('Help with program'); HIV Synthesis | HIV Modelling.

FSWs are explicitly modelled within the model; this has been extensively described elsewhere[8]. In short, all women in the model have a lifetime susceptibility to initiating sex work classed as low, medium or high, reflecting different social circumstances. For women aged 15–49, their lifetime susceptibility, along with age, overall population levels of risk behaviour and whether they have previously been a sex worker determines their likelihood of becoming a FSW in any one 3 month period. In any 3 month period in which a woman is considered a FSW, she may not be continually active as a FSW. Hence, she may have zero short-term condomless partners in any given 3 month period due to either being inactive, or using condoms consistently. Without any programme in place FSW are assumed to have lower access to support and health services than other women[12,13]. These disadvantages have been captured in the model by assuming FSW not attending a programme are more likely to interrupt treatment, more likely to be lost at diagnosis and less likely to be adherent to treatment, resulting in lower rates of viral suppression. We use wide sampling frames to reflect the uncertainty over these disadvantages[8] (Supplementary material, page 90).

The model was calibrated to the HIV epidemic in Zimbabwe using publicly available data sources including the Demographic Health Surveys (DHS) and the Population Based HIV Assessments (PHIAs)[14–17]. Of 2000 runs, 100 were selected as having good calibration. For each run, parameters were selected from random distributions[8]. The resulting uncertainty is expressed as 90% intervals for each modelled outcome. Key outputs of the calibration, including number of people

living with HIV, prevalence, incidence and the proportion of PLHIV diagnosed, on ART and virally suppressed are shown in the Appendix. On the whole, modelled data aligned well with empirical data (Table A1 and Table 1). Further details of calibration to Zimbabwe have been previously made available[18].

### The KP programme (usual care)
The KP programme has provided free HIV and sexual health services to FSWs since 2010 across Zimbabwe. The programme is supported by a network of trained peer educators, and services include provision of contraception, HIV testing, provision of PrEP and referral for HIV-positive women to public sector clinics for antiretroviral therapy (ART). Outreach worker supervisors (ORWs) meet with peer educators (also known as empowerment workers) once a month to support and counsel FSW[4].

We modelled a sex worker programme intended to represent the KP programme from 2010 onwards. The programme was assumed to have a positive impact on condom use (number of condomless partners reduced by two thirds in up to 10% of sex workers selected at random), HIV testing (6-monthly HIV testing in up to 50% of sex workers), PrEP uptake from 2019 onwards (willingness to take PrEP increased to up to 10% in sex workers) and long term curable STIs (chances of persistence of a curable STI lasting beyond 3 months reduced by up to 20%)[6,19]. We further assumed any disadvantages FSW had were reduced, negated and/or resulted in an advantage for FSW. Specifically, we assumed FSW living with HIV who attended the programme were less likely to interrupt ART treatment or be lost at diagnosis and were more likely to have good adherence to treatment. We allowed for varying levels of coverage of the programme to reflect the potential variation of the nationally scaled programme across Zimbabwe. Parameters reflecting the impact of the programme are shown in the Table 5. We assumed up to a 10% probability of entering the programme per 3 months and default of continued engagement thereafter but with a probability of discontinuing care of up to 10% per 3 month period.

### The AMETHIST intervention
The AMETHIST trial was a cluster randomised trial nested within the KP programme[7]. Twenty-two clusters (FSW population around dedicated clinics) were 1:1 randomly allocated to usual care as described above,

**Table 5 | Effects of the Key Population (KP) programme and additional effects of the AMETHIST intervention**

| Per 3 months | Probabilities sampled with equal likelihood according to sex work programme | |
| --- | --- | --- |
| | KP (assumed since 2010) | AMETHIST* (additional to KP from 2024) |
| Engagement with programme | 1%, 3%, 5% 10% | 10% increase compared to KP |
| Disengagement from programme | 2%, 5%, 10% | 10% decrease compared to KP |
| Greater condom use (condomless partners reduced by two-thirds) | 5%, 10% | No additional effect |
| 6-monthly testing | 20%, 35%, 50% | No additional effect |
| Increase in willingness to take PrEP | 5%, 10% | No additional effect |
| Persistent STI reduction | 10%, 20% | No additional effect |
| In relation to disadvantages: | | |
| Interruption disadvantage reduced by: | 30%, 50%, 70% | Disadvantage reduced further by up to 3-fold compared to KP |
| Loss at diagnosis disadvantage reduced by: | 30%, 50%, 70% | Disadvantage reduced further by up to 3-fold compared to KP |
| Adherence disadvantage reduced as follows: Original adherence + (1 – original adherence * (sampled probability)): | Sampled probability: 10%, 15%, 25% | Disadvantage reduced further by up to 4-fold compared to KP |

*Additional AMETHIST impact was guided by results of the trial[1], shown in the last column of Table 2.

or to *AMETHIST* interventions. The AMETHIST intervention was implemented in addition to the KP programme and comprised peer-led microplanning and self-help groups among sex workers. The role of the peer-led microplanners was to provide risk-differentiated support for all sex workers in a given geographic hotspot, determined by their level of vulnerability which was assessed by the peer micro-planners every three months. Sex workers perceived to be at high risk were seen weekly, whilst those at lower risk were seen less often. All support provided was HIV status neutral. Microplanners met with ORW each week to review their caseload and discuss any challenges with providing risk-differentiated support. A sub-set of sex workers were invited to engage in self-help groups which aimed to build social cohesion amongst FSW and develop psychological and potentially financial resilience amongst members.

The AMETHIST intervention was found to have a significant impact on FSW living with HIV, though not on HIV negative FSW who were at risk of acquisition of HIV. Specifically, 93.5% FSW living with HIV in the AMETHIST arm compared to 88.8% with HIV in the usual care arm had undetectable viral loads at the end of the study period (−5·5% (95% CI −8·2% to −2·9%)). Hence, when modelling the intervention, we assumed the 'AMETHIST' programme would have no impact on prevention of HIV but would have benefits compared with the KP programme alone for FSW living with HIV in relation to treatment interruption, being lost at diagnosis and ART adherence. The magnitude of these benefits are shown in Table 5.

For each 'scenario'/model run, we compared outcomes from (i) continued implementation of the KP programme to (ii) continued implementation of the KP programme + the AMETHIST intervention from 2024 onwards. We assumed for AMETHIST a 10% higher likelihood of being seen at the programme than with the KP programme alone reflecting the higher number of FSW seen with the intervention[7]; we expect this to be a conservative estimate given AMETHIST took place during the COVID-19 lockdowns.

The cost of the KP programme was estimated at $132 per year per sex worker seen throughout the year, whilst the cost of the AMETHIST programme was estimated at $155 per year per SW seen (i.e., AMETHIST itself incurs an additional cost of $23 per year per sex worker) (manuscript in preparation). This was based on a detailed prospective economic evaluation which measured full costs of scaling up AMETHIST from the provider perspective and following international costing guidelines. In this economic evaluation top-down financial expenditure analysis and categorisation by input type was layered on a site level bottom-up micro-costing exercise where service provision and resource input use was quantified. Disability adjusted life years (DALYs) for the whole adult population were calculated and

compared across the two programmes. Similarly, projected total costs for HIV incurred by the health care system in Zimbabwe were also compared across the two programmes. The net benefits, in terms of DALYs averted, by AMETHIST compared to the standard of care KP program were calculated as the incremental DALYs averted by AMETHIST over the standard of care KP programme minus the incremental costs of AMETHIST divided by the assumed cost-effectiveness threshold of US$500 per DALY averted.

The cost-effectiveness analysis was conducted from a healthcare perspective, costs and health outcomes were both discounted to present US$ values at 3% per annum.

Similar analyses were conducted comparing continuation of the KP programme to discontinuation of the programme from 2024 onwards in response to the funding cuts.

### Reporting summary
Further information on research design is available in the Nature Portfolio Reporting Summary linked to this article.

## Data availability
This modelling study is based on simulations and there is no analysis of empirical data. Model parameters are included in the Supplementary Information.

## Code availability
The code is available on Figshare: HIV Synthesis program files for AMETHIST analyses

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

## Acknowledgements

This study was made possible by an award from Wellcome Trust (Grant number: 214280/Z/18/Z). The funders of the study had no role in study design, data analysis, data interpretation, or writing of the report.

## Author contributions

Contributions to conception of the modelling question: F.C., A.P., L.B.M., J.H.; Running the models/ model development and coding: L.B.M., A.P.; Drafting of initial manuscript: L.B.M.; Critical input into draft manuscript: C.M., P.R., F.M., S.A., S.T.C., P.M., R.S., A.M., O.M., I.T., J.H., F.C., A.P.; L.B.M. and A.N.P. accessed and verified the model program and output data. All authors had full access to all the data in the study and had final responsibility for the decision to submit for publication

## Competing interests

The authors declare no competing interests.
