## [Transparent Peer Review file · Nature Communications]

Impact and cost-effectiveness of the community-led AMETHIST intervention among female sex workers in Zimbabwe

Corresponding Author: Dr Loveleen Bansi-Matharu

Version 0:

Reviewer comments:

Reviewer #1

(Remarks to the Author)

This modeling study evaluates the long-term impact and cost-effectiveness of the AMETHIST intervention and Zimbabwe's Key Populations (KP) programme for female sex workers (FSW) using the HIV Synthesis Model. The results suggest that AMETHIST improves HIV outcomes and is cost-saving compared to the KP programme alone, while discontinuing the KP programme leads to worse outcomes and higher costs. However, the study's conclusions rely heavily on model assumptions and parameter choices, which require clearer justification and documentation. The manuscript in its current form reads more like a condensed article suitable for a specialized HIV journal than a stand-alone study appropriate for the broad and interdisciplinary readership of Nature Communications.

The manuscript lacks the methodological detail necessary for evaluation by experts in agent-based modeling. As currently presented, the modeling framework is neither sufficiently described nor reproducible. For those with a technical background who are capable of assessing the model, the current documentation falls short of what is needed to verify the model's structure and assumptions. I was unable to assess the technical correctness of the model components and would have to rely solely on the assertion that the baseline model was previously published multiple times.

Although the authors have made their code publicly available, I did not find it to be sufficiently documented or reproducible to the standards expected by Nature Communications. The model outputs depend heavily on assumptions and parameterizations that remain unclear, limiting the reader's ability to interpret or trust the scenarios presented. The authors could make a valuable contribution to the modeling community by ensuring their model is both transparent and reproducible, rather than referring the reader to prior publications of the HIV Synthesis model, many of which, btw, focused on countries other than Zimbabwe.

Abstract

The abstract is not well suited for the broad readership of Nature Communications. Several terms and references (e.g., "USAID Stop-Work order") are too specific to be understood by readers outside the HIV policy field and should be revised for clarity and accessibility. Costs are reported both in pounds and US dollars (typo?).

Title

The title mentions the general population HIV incidence but the results in the abstract and the text mostly concern FSW.

Model calibration: fitting to one data point

The authors present results of model fitting for the general population in Table A0, which appears to reflect calibration to a single year of data. For any model, particularly one of this complexity, this is insufficient and unreliable. A figure showing the model fit to all data used in calibration (as indicated in Table A0 or elsewhere in the supplement and main text) should be included to allow proper assessment. In Table 2 in the main text, no observed metrics for the general population are reported; moreover, once again, comparisons are made for a specific year/one time point. With the free parameters this model has, I don't find this procedure reliable at all.

Model validation and risk of overfitting

There is no discussion or presentation of model validation against independent data not used in the fitting process. Without validation, and given the model's complexity, the risk of overfitting is high and cannot be assessed. Validation figures should be included alongside model fitting figures.

Sensitivity analyses

Sensitivity analyses for estimated or fixed parameters are not presented. These are essential to evaluate the robustness of the model results and should be added.

Data sources

Are the authors fitting the model to the FSW-specific data and also the general population data? The sources of data used for model fitting for HIV outcomes in the general population are not clearly described in one place, and the references are inconsistently cited. For instance, Table A0 includes a footnote referencing a 2017 report, while the table lists 2020 data. It is unclear why incidence data from UNAIDS are not used, and inconsistencies between sources are not explained. Additionally, Table A0 appears to be missing footnote 1.

Parameter estimates and supplement clarity

The supplementary material is lengthy but lacks clarity. It is not clear which parameters were calibrated, estimated, or fixed; all this information is scattered. These distinctions should be summarized in a dedicated table. Furthermore, (prior and) posterior parameter estimates are not provided; a summary table would improve transparency.

Outdated literature and reference errors

The literature supporting model parameterization is often outdated, particularly concerning ART drug resistance (with exception of recent papers for LA-PrEP) and cost estimates. Several citations are incorrectly formatted or missing, including "Agegnehu 2022" (possibly a typo), "Hakim at el. Mfinanga et al.", "Irvine et al.", "Massud et al.", and "Mishra".

Geographic specificity of data and references

There are few Zimbabwe-specific references, and data from other countries are used without justification. A clear distinction should be made between country-specific and external sources, especially for parameters or priors. This could be addressed by including a table or discussion highlighting which data sources are applicable to Zimbabwe and which are extrapolated.

Code documentation and reproducibility

The code is available, but poorly documented and not reproducible. There is no README; there is just a very short description with each of the 3 files uploaded (the link mentioned in the Code and Software Submission Checklist points to one of them). Moreover, the files are not cleaned, and there is a lot of commented code. I had great difficulty figuring out how to run it.

Fitting procedure and reproducibility

The fitting procedure is not reproducible. I cannot find any file with the fitted parameters, with the data used. A random number seed is not set, as the authors state themselves: "This is a stochastic model and hence running the model 2000 times will not produce identical outputs to those analysed." That is correct, but for the readers to have confidence in the reproducibility of results, a random seed could be fixed so that the model can be run and outputs compared to what is reported in the paper (but that information is lacking too, as I described above).

Scenarios

With regards to scenarios, my compliments on considering the reduction of funding for FSW programmes. It is very important that this scenario was included in the analyses.

(Remarks on code availability)

The code, in its current form, is not usable by the community. Despite investing considerable time, I was unable to run it, which I found quite frustrating. Researchers who are genuinely interested in engaging with the model would likely face similar difficulties. I have included some specific comments in my review, but I would strongly recommend that the authors consult recent papers in Nature Communications that provide well-documented and accessible code as examples of good practice.

Reviewer #2

(Remarks to the Author)

This manuscript investigates the important topics of the potential long-term impacts and costs of scaling up the "AMETHIST" intervention for SW in Zimbabwe, continuing the current KP program without "AMETHIST," and discontinuing the KP program. This investigation is timely given the recent primary findings of the AMETHIST trial and the recent USAID dismantling.

The manuscript uses the "Synthesis" model to perform its investigation, which is a well-established thoroughly reported model for investigating HIV-related topics in the region. The manuscript is clearly presented, and the conclusions flow clearly from the results.

I have no major methodological critiques for the approach at this time. I do have a number of questions or comments relating to adding clarity to the approach, the findings, or their presentation, and I believe it is worthwhile that they be addressed

before the manuscript is accepted for publication. It is my expectation that they can be relatively easily addressed, barring surprises I imagine the manuscript should be acceptable afterwards.

My comments and questions are as follows, generally in sequence with the manuscript.

- Abstract:

Results: The results note that both AMETHIST vs. KP program only and the KP program only vs discontinuation were cost-saving. It would be very informative to clarify when within the 50-year time horizon cost savings were achieved.

- Introduction:

Line 114-115: Please share the magnitude and uncertainty (CI or p-value) of this risk reduction, as it is important context for the reader.

Line 117-118: Please share by what definition / threshold is being used when stating the KP program has been operating "cost-effectively"

Methods:

255: The authors have (I assume inadvertently) placed the methods section at the end. It should be shifted to in between Intro and Results.

277 - 280: The logic of this cascade is reasonable. Further, it is appreciated that Table S27 is quite extensive and understood that within-manuscript space is always limited in available word counts to truly detail a full model. That said, changes to FSW-related parameters are the key experimental inputs for this investigation of predicted KP program effects, so I feel these disadvantage parameters deserve quantification and sourcing / justification within the main text. I believe either narrative or tabular form would be acceptable. If tabular, something akin to table A1 would be appropriate.

283: Please spell out the acronym PHIA in its first usage

285-287: Looking at A0, it looks like calibration was based on general population parameters. Were any FSW-specific indicators used to assess calibration? Overall, the calibration looks reasonable, but if no FSW-specific indicators were utilized, I would, either a) ideally check and report in A0 that they are also well calibrated, or b) discuss (here) as a limitation of the calibration that they either weren't available or did not calibrate as well as the general indicators in A0. [I believe this does get briefly mentioned later with respect to incidence, but further and earlier clarity would be appropriate.]

290 - 296: Consider placing this paragraph in the Introduction rather than Methods section.

303 - 305: I assume this refers specifically to the +AMETHIST; scenario. If so, please make that explicit.

309: Similar to my comment above, I'd put A1 or a version thereof into the main text near here. Adding / removing those specific parameters to the scenario runs is the experiment under investigation, and thereby warrant main text presentation.

Regarding A1, wherever placed, the reference to Cowan 2024 for the third column is much appreciated. To the extent that the second column (regarding KP program related parameters) relies on prior literature vs. assumption / broad sampling, any relevant citations should also be clearly noted.

328 - 329: This manuscript would be strengthened by also providing the reported confidence interval around or a p-value for this difference.

352: Please clarify that it is "projected" total costs that are compared

353-359: This phrasing could be cleaned up. Nested parentheses are fine in math but a bad sign in writing, but more importantly it describes a net benefits calculation (in DALYs) but confusingly refers to it as calculating incremental DALYs informing a cost-effectiveness calculation, when in fact a cost-effectiveness ICER is never actually calculated.

Results:

162 - 163 & 173: The authors note that continuing the KP program is cost-saving compared to discontinuation within 20 years, and adding AMETHIST saves both costs and DALYs within 15 years. I think it is very worthwhile to specify specific years in which those achievements are realized. Both present-bias and genuine credit constraints can inform whether long-term positive returns will or even can be pursued by various decision-makers. With respect to USAID in particular, US assistance has for the short-run (and possibly longer) theoretically lended to treatment and at abdicated prevention. If available, reporting what year treatment costs alone under discontinuation surpass the combined treatment and prevention costs of continuing the KP program would be the most politically pertinent information to present to stakeholders.

- DISCUSSION

200: Please re-phrase "would" eliminate issues to "could" eliminate issues.

200-208: While it is indeed important to report that this analysis has not accounted for recently proven long-acting injectable PrEP formulations, the subsequent rush through related research by related authors and extrapolating projections therefrom strays somewhat afield from the evidence generated in this investigation. I would summarize into a more brief statement along the lines that that cost effectiveness research on injectables implies that it is worth assessing under what circumstances and timelines they too might be cost-saving.

226-235: Limitations of the primary AMETHIST analysis itself should also be acknowledged within these limitations, particularly that they are based on endline group comparisons absent baseline balance confirmation.

Table 1: The phrasing "Percentage of SW with total duration in each category (years)" is unintuitive. Perhaps say, "Percentage of SW with total years spent as SW"?

Table 2: Does "Portion of SW tested in past year" refer to all SW, all HIV- SW, or all SW considered "At risk"? Also "Proportion of SW diagnosed" should be clarified to "Proportion of SW living with HIV diagnosed"

Table 3: I don't see lines referring to Non_ART programme costs, but they seem included to achieve \$149M & \$147M, correct? I recommend making at least one line for those.

Supplementary Material:

I'd request that the Table of Contents for the Model Details section of the Appendix (p 5-6) further enumerate major headings within Table S27: Distribution of Parameters. For now there are 20+ pages of parameters to scan in search of the ones a reader wishes to delve more into.

Table S29: Multiple of these unit costs come from 2014 papers. Please discuss if / how they were updated to reflect 2024 unit costs.

(Remarks on code availability)

Version 1:

Reviewer comments:

Reviewer #1

(Remarks to the Author)

I am grateful to the reviewers for their excellent rebuttal and for sharing the model documentation on the dedicated webpage. This resource will be valuable for the community interested in exploring the model in greater depth.

I do, however, have a remaining concern regarding the decision not to fix the random seed. The authors responded:

"As such, each research question involves hundreds of parameters selected at random for each run. The model for this analysis was run 2000 times, and outputs across the HIV epidemic were considered suitable for inclusion in 100 of these runs, i.e. the calibration process. By setting a seed, we would be hugely restricting the uncertainty that we would like to convey across epidemic outputs. Instead, to reproduce the results, we would suggest running the model 2000 times as was originally done for these analyses, and analysing the approximately 100 runs that would pass the fit criteria coded in the core file and refined in the 'create wide file'. We feel this would be much more useful than producing any one run using a set seed."

I would like to clarify that my request was not about removing stochasticity from the model, which I fully recognize as essential in individual-based models. Rather, my concern relates to the reproducibility of the specific results presented in the manuscript. Running the model on a different computer or environment (e.g., a different SAS installation) could yield non-identical outputs, as random number generation may depend on local factors such as system time.

To address this, one possible approach would be to fix the initial random seed (from which the first number of a sequence is generated) to ensure reproducibility of reported results. Alternatively, comparable techniques could be applied to achieve the same goal. This would not constrain the stochasticity of the model overall, but it would allow independent researchers to replicate the precise outputs underlying the figures and analyses in the manuscript.

(Remarks on code availability)

Documentation and commenting of the model code have been much improved.

Reviewer #2

(Remarks to the Author)

In this revised manuscript, the authors have done well at mostly addressing my previous comments and concerns. I have one remaining key request and a few minor edits.

The key request, as per my previous comments, is within-text presentation of the key parameters regarding FSW and how they are changed under different scenarios. As the parameters which vary between scenarios, they constitute the experiment being investigated and their presentation is important to conveying to the reader the nature of the investigation. These can be tabular and / or narrative, but must be within the main text. This information will come primarily from Table A1, but also the "parameters related to sex worker disadvantages" and "Parameters related to sex worker programs" sections of S27 on p 115 & 116 of the supplemental appendix. NOTE that A1 and s27, whilst concordant on most parameters, have *different values* for the effect of the KP program on adherence disadvantage reductions, being .1, .15, .25 in A1 but .2, .35, .5 in S27. These must be aligned with each other and the code as implemented. Relatedly, my understanding of the parameters as implemented is that there is discordance with their numeric presentations in A1 and S27 and the claim that on lines 307-309 that disadvantages were reduced, "negated, and/or resulted in an advantage" given that the multiplicative proportions in A1 and S27, applied to any disadvantage, cannot fully reach zero or change sign. These apparent discrepancies partially undergird my insistence that these parameters receive clear in-text attention.

Otherwise, I feel my concerns have been addressed, with the following few and minor changes requested to finalize their editing.

1) I'd previously asked by what definition / threshold the authors were using the term "cost-effectively" on line 117. The authors have clarified that their definition was that utilized in reference 8. Looking at reference 8, I see the standard there is \$500/DALY averted. I request that the authors incorporate this benchmark (\$500/DALY) within the sentence describing the KP program as cost effective.

2) Line 285 points readers to A0 for evidence that modeled data calibrated well to empirical data. I see now that readers should be pointed to both A0 AND Table 1. The additional reference to further details in a differing publication is also appreciated, though the reference needs formatting.

3) I appreciate the authors' updating of the phrasing of lines 357-360. It is improved but still a bit incongruent with standard economic terminology by conflating net and incremental benefits. I would phrase this part as "The net benefits, in terms of DALYs averted, by AMETHIST compared to the standard of care KP program were calculated as the incremental DALYs averted by AMETHIST over the standard of care KP programme minus the incremental costs of AMETHIST divided by the assumed cost-effectiveness threshold of US\$500 per DALY averted."

(Remarks on code availability)

We thank the reviewers for their comments and have added responses to each comment in green.

Please note that in response to one of the reviewer's comments, and given the current status of injectable PrEP, we have rerun the model now including injectable PrEP as an additional mode of PrEP to oral PrEP. Whilst numbers have slightly changed, conclusions remain the same as the previous version.

REVIEWER COMMENTS

Reviewer #1 (Remarks to the Author):

This modeling study evaluates the long-term impact and cost-effectiveness of the AMETHIST intervention and Zimbabwe's Key Populations (KP) programme for female sex workers (FSW) using the HIV Synthesis Model. The results suggest that AMETHIST improves HIV outcomes and is cost-saving compared to the KP programme alone, while discontinuing the KP programme leads to worse outcomes and higher costs. However, the study's conclusions rely heavily on model assumptions and parameter choices, which require clearer justification and documentation. The manuscript in its current form reads more like a condensed article suitable for a specialized HIV journal than a stand-alone study appropriate for the broad and interdisciplinary readership of Nature Communications.

The manuscript lacks the methodological detail necessary for evaluation by experts in agent-based modeling. As currently presented, the modeling framework is neither sufficiently described nor reproducible. For those with a technical background who are capable of assessing the model, the current documentation falls short of what is needed to verify the model's structure and assumptions. I was unable to assess the technical correctness of the model components and would have to rely solely on the assertion that the baseline model was previously published multiple times.

Although the authors have made their code publicly available, I did not find it to be sufficiently documented or reproducible to the standards expected by Nature Communications. The model outputs depend heavily on assumptions and parameterizations that remain unclear, limiting the reader's ability to interpret or trust the scenarios presented. The authors could make a valuable contribution to the modeling community by ensuring their model is both transparent and reproducible, rather than referring the reader to prior publications of the HIV Synthesis model, many of which, btw, focused on countries other than Zimbabwe.

As the reviewer notes, the model used for these analyses has been made publicly available on Figshare and the baseline model has been published many times. We

do agree that it is not a straightforward task to understand all of the modelling code; for most users this is a long process with interaction from model developers. However, the outputs from the model are reproducible, allowing for stochastic variation. The files used in this analysis were uploaded to Figshare, each accompanied by a brief summary describing its purpose. In response to this review, these summaries now contain considerably more detail, and the comments in the code have been refined. We have also added additional clarification comments to the programs. New files can be found here: HIV Synthesis program files for AMETHIST analyses

We did also include a supplementary file containing a detailed description of the model (Appendix, starting at page 5) that we feel addresses the reviewer's concerns – in particular concerns around justification of assumptions, parameters and overall components of the model. In line with Nature Communication publications, we have written this manuscript in a manner such that it is interpretable by a wide readership and hence have not focussed on the technical details of the model within the body of the paper. We have now included a sentence directing readers to the model details in the first paragraph of the Methods section. In response to the reviewer's concerns, we have now included additional support for users interested in running the model code. This support is now mentioned both within the main paper (at the end of the first paragraph of the Methods) and in the supplementary material. Specifically, we have added a slideset titled "*Help with Program*", which provides further explanation of the model code. We have inserted a sentence at the beginning of the model details in the supplementary material to guide readers to these resources: "*In addition to the details provided below, we encourage anyone wishing to run the model to visit: HIV Synthesis | HIV Modelling, and in particular, to view the slideset titled 'Help with Program'.*"

Abstract

The abstract is not well suited for the broad readership of Nature Communications. Several terms and references (e.g., "USAID Stop-Work order") are too specific to be understood by readers outside the HIV policy field and should be revised for clarity and accessibility. Costs are reported both in pounds and US dollars (typo?).

We have changed the language in the abstract, and throughout so it is less specific, and clarification is provided where needed. The typo using £s instead of \$s has been corrected.

Title

The title mentions the general population HIV incidence but the results in the abstract and the text mostly concern FSW.

We have edited the title so it now reads: Impact on **female sex workers** and general population HIV incidence and cost-effectiveness of the enhanced AMETHIST

intervention for sex workers in the context of an existing sex worker programme in Zimbabwe

Model calibration: fitting to one data point

The authors present results of model fitting for the general population in Table A0, which appears to reflect calibration to a single year of data. For any model, particularly one of this complexity, this is insufficient and unreliable. A figure showing the model fit to all data used in calibration (as indicated in Table A0 or elsewhere in the supplement and main text) should be included to allow proper assessment. In Table 2 in the main text, no observed metrics for the general population are reported; moreover, once again, comparisons are made for a specific year/one time point. With the free parameters this model has, I don't find this procedure reliable at all.

Table A0 was intended to give the reader an idea of how close the modelled outputs were to the most recent empirical data. As the reviewer notes, this is a complex model and model outputs are compared to a wide range of data. We did not include figures showing this in the submission as such figures would add substantially to the manuscript. We have however now added a series of graphs to the Appendix showing trends over time of the key outputs in line with empirical data and added a reference to the calibration in the Methods section when first discussed.

Table 2 shows model projections for 2030. We have not added observed data for the general population here as it would not differ from that shown in Table A0, i.e. the most recent data available. Further, we feel comparing current empirical data to projected modelled data in 2030 for the general population is not appropriate. We did add data from the trial to this table to give the reader an idea of how close our modelled projections were to those specifically seen in the trial as this is the focus of this paper. Data in the trial were collected at a single point, but we do not agree that this impacts on reliability. The comparison was not intended to be a direct comparison between modelled and observed data or a measure of calibration; projected outputs are for 2030 and observed data was collected in 2021. Rather, we included the trial data here as a means of the reader having knowledge of the impact of the trial without having to look at the references and be able to indirectly compare our modelled future projections with the trial's current findings.

Model validation and risk of overfitting

There is no discussion or presentation of model validation against independent data not used in the fitting process. Without validation, and given the model's complexity, the risk of overfitting is high and cannot be assessed. Validation figures should be included alongside model fitting figures.

In the model details, we discuss the approach we have taken to developing the model. In general, we have not 'fitted' to observed data by inputting specific values for key epidemic characteristics such as prevalence or incidence. Instead, these are considered 'outputs' of the model and have been assessed by looking across a wide range of outputs to ensure modelled data is triangulated to the many empirical data

points that are available. We feel overfitting is not a cause for concern in this case as the model validation is almost always against independent data. Further, we have used all available observed data on FSW for this exercise. Further details of the model fit can be found in the newly referenced paper (HIV Incidence and Prevalence Projections for Zimbabwe: Findings from Five Mathematical Models by Isaac Taramusi, John Stover, Daniel Citron, Michael Pickles, Debra ten Brink, Tsitsi Apollo, Loveleen Bansi-Matharu, Anna Bershtyn, Robert Glaubius, Jeffrey Imai Eaton, Rowan Martin-Hughes, Amon Mpofu, Owen Mugurungi, Getrude Ncube, Paul Revill, Ngwarai Sithole, Richard Makurumidze, Simbarashe Rusakaniko, Jennifer Smith, Andrew N. Phillips, Valentina Cambiano :: SSRN). We have also added calibration graphs to the Appendix showing model fit against observed data and agree with the reviewer that these will help the reader assess the calibration.

Sensitivity analyses

Sensitivity analyses for estimated or fixed parameters are not presented. These are essential to evaluate the robustness of the model results and should be added.

Each run of the model is generated by sampling across parameter values to reflect uncertainty in assumptions (see third paragraph of the Methods and table S27 in the Appendix). This parameter (and stochastic) uncertainty is presented in the form of 90% ranges for all modelled outputs. We have now added to the sentence in the Methods to clarify this: *'For each run, parameters were selected from random distributions(8). The resulting uncertainty (and stochastic uncertainty) is expressed as 90% ranges for each modelled outcome.'*

Data sources

Are the authors fitting the model to the FSW-specific data and also the general population data? The sources of data used for model fitting for HIV outcomes in the general population are not clearly described in one place, and the references are inconsistently cited. For instance, Table A0 includes a footnote referencing a 2017 report, while the table lists 2020 data. It is unclear why incidence data from UNAIDS are not used, and inconsistencies between sources are not explained. Additionally, Table A0 appears to be missing footnote 1.

Our modelled outputs are compared to both FSW-specific data (Table1) and data from the population as a whole to assess model fit (Table A0). The PHIA reference cited in Table A0 was an error, this should have been the 2020 report rather than the earlier 2017 report – this has now been corrected. Table A0 has been reformatted to make clearer where the observed data are from, additional items have been added including the UNAIDS incidence estimate (although we note that these are model estimates themselves and not strictly observed data). We do not consider there are inconsistencies between the modelled data and the observed data; on the whole, observed data is within the 90% range of the modelled data. The PHIA data are considered high quality nationally representative data, whilst UNAIDS data are model estimates; we have included both in the comparison for completeness.

Parameter estimates and supplement clarity

The supplementary material is lengthy but lacks clarity. It is not clear which parameters were calibrated, estimated, or fixed; all this information is scattered. These distinctions should be summarized in a dedicated table. Furthermore, (prior and) posterior parameter estimates are not provided; a summary table would improve transparency.

A summary table is included at the end of the supplementary material, page 89, detailing parameter distributions. The vast majority of parameters are selected at random from distributions that reflect the epidemic across East, Central, South and West Africa - those that aren't are specified in the table. For this analysis, we overwrite some parameters to fit specifically to the Zimbabwe epidemic. We agree with this reviewer that this wasn't sufficiently clear and have now added a note to the summary table where this is the case. Whilst we do select parameters from random distributions to reflect the epidemic in any given setting, we do not specifically input epidemic characteristics such as HIV prevalence or incidence; these are outputs of the model.

Although the prior distributions are provided in Table S27 of the Appendix, the specific parameter values used in the analysis—those selected for their good fit to the particular epidemic—have not been presented in this paper, nor has this approach been adopted in previous publications. The reason for this is that we do not feel that adding these distributions would be particularly helpful to the reader, particularly beyond the tables comparing observed and modelled data. We have however now added graphs showing key modelled outputs against observed data over time in the Appendix – we feel this addresses any concerns the reviewer may have regarding the calibration procedure.

We have also added some additional text to the supplementary material clarifying the parameter selection process, reading as follows on Page 111, as a footnote to the summary table: *Parameters are randomly chosen from distributions as shown in the table above. The value of these parameters will impact on key epidemic characteristics such as HIV prevalence, incidence, proportion diagnosed, on ART and virally suppressed. The latter are all outputs of the model and as such are not directly inputted into the model.*

Outdated literature and reference errors

The literature supporting model parameterization is often outdated, particularly concerning ART drug resistance (with exception of recent papers for LA-PrEP) and cost estimates. Several citations are incorrectly formatted or missing, including “Agegnehu 2022” (possibly a typo), “Hakim at el. Mfinanga et al.”, “Irvine et al.; Massud et al.”, and “Mishra”.

The literature supporting parametrisation is updated when a meaningful difference or a shift in effect size is evidenced. Hence, the reviewer is correct that many of the references are not recent as what we know about drug resistance to specific drugs was indeed learned some time ago and there are not more recent papers which add

substantively to those insights – much like the natural history of HIV. We feel that this is not an issue – particularly since the vast majority of parameters are selected from random distributions allowing for uncertainty. However, we are happy to reconsider our parameter distributions for any specific parameters that the reviewer feels we have not adequately captured as a result of more recent findings.

We have now checked through the citations and reformatted and corrected those which were incorrect.

Geographic specificity of data and references

There are few Zimbabwe-specific references, and data from other countries are used without justification. A clear distinction should be made between country-specific and external sources, especially for parameters or priors. This could be addressed by including a table or discussion highlighting which data sources are applicable to Zimbabwe and which are extrapolated.

We acknowledge the importance of parameter choices in local data wherever possible. However, in some cases, Zimbabwe-specific data were not available or were limited in scope, particularly for certain epidemiological and behavioural parameters. In such instances, we drew on data from the broader sub-Saharan African region. These selections were informed by existing literature suggesting regional similarities in key indicators and resulting outputs align well with observed data in Zimbabwe (Table A1).

As mentioned in a previous response, we have now distinguished between Zimbabwe specific parameters and those that are region wide. We have shown in Table 1 how closely our outputs for FSW align with observed data in Zimbabwe; these modelled outputs are a result of the parameter distributions specific to Zimbabwe. References for the observed data have been included in Table 1 (generally the data from the AMETHIST trial in Zimbabwe).

Code documentation and reproducibility

The code is available, but poorly documented and not reproducible. There is no README; there is just a very short description with each of the 3 files uploaded (the link mentioned in the Code and Software Submission Checklist points to one of them). Moreover, the files are not cleaned, and there is a lot of commented code. I had great difficulty figuring out how to run it.

As the reviewer notes, this code is complex and we had left comments in to aid interpretation but agree some comments ought to be removed. This has now been done, and three 'clean' files have been uploaded to Figshare. Results are reproducible, allowing for stochastic variation. The model is coded in SAS and to run the model, the 'core' file would need to be run multiple times to allow users to create output files from which summary measures for key epidemic characteristics can be produced using the 'create wide file' and 'analysis file'. As mentioned in an earlier

response, we have now also included a slideset which provides further explanation of the model code and help on how to run it.

Fitting procedure and reproducibility

The fitting procedure is not reproducible. I cannot find any file with the fitted parameters, with the data used. A random number seed is not set, as the authors state themselves: "This is a stochastic model and hence running the model 2000 times will not produce identical outputs to those analysed." That is correct, but for the readers to have confidence in the reproducibility of results, a random seed could be fixed so that the model can be run and outputs compared to what is reported in the paper (but that information is lacking too, as I described above).

The HIV Synthesis model is a well-established and extensively validated individual-based model of HIV. It is recognized as one of the more sophisticated modelling frameworks currently in use and has played a significant role in informing HIV policy. Research utilizing the HIV Synthesis model has been widely published in peer-reviewed journals. As such, each research question involves hundreds of parameters selected at random for each run. The model for this analysis was run 2000 times, and outputs across the HIV epidemic were considered suitable for inclusion in 100 of these runs, i.e. the calibration process. By setting a seed, we would be hugely restricting the uncertainty that we would like to convey across epidemic outputs. Instead, to reproduce the results, we would suggest running the model 2000 times as was originally done for these analyses, and analysing the approximately 100 runs that would pass the fit criteria coded in the core file and refined in the 'create wide file'. We feel this would be much more useful than producing any one run using a set seed.

Scenarios

With regards to scenarios, my compliments on considering the reduction of funding for FSW programmes. It is very important that this scenario was included in the analyses.

Thank-you, we agree it is essential to consider these results given the current funding landscape.

Reviewer #1 (Remarks on code availability):

The code, in its current form, is not usable by the community. Despite investing considerable time, I was unable to run it, which I found quite frustrating. Researchers who are genuinely interested in engaging with the model would likely face similar difficulties. I have included some specific comments in my review, but I would strongly recommend that the authors consult recent papers in Nature Communications that provide well-documented and accessible code as examples of good practice.

We thank the reviewer for diving deep into the code and apologise for any frustration caused. We are surprised that if the Reviewer has SAS installed that the code would not simply run - there are no input data files. We have however made the instructions on Figshare more detailed to help aid in the running of the code.

The model is detailed and complex; understanding all the SAS code is many months if not years of work but it should not be necessary to simply run the code multiple times and thus reproduce the results. The core model itself is run multiple times using dedicated computing nodes. It is possible to run the model on a high specification computer but this may take considerable time. Whilst we encourage anyone interested in the code to get in touch with the modelling team, we would like to place emphasis on readers comparing the modelled data to observed data to judge calibration.

As mentioned in an earlier response, in light of the reviewers concerns, we have now also included a series of graphs showing modelled data and observed data specifically for Zimbabwe. These graphs illustrate parameter and stochastic uncertainty across a range of outputs and we feel address many of the reviewer's concerns.

Reviewer #2 (Remarks to the Author):

This manuscript investigates the important topics of the potential long-term impacts and costs of scaling up the "AMETHIST" intervention for SW in Zimbabwe, continuing the current KP program without "AMETHIST," and discontinuing the KP program. This investigation is timely given the recent primary findings of the AMETHIST trial and the recent USAID dismantling.

The manuscript uses the "Synthesis" model to perform its investigation, which is a well-established thoroughly reported model for investigating HIV-related topics in the region. The manuscript is clearly presented, and the conclusions flow clearly from the results.

I have no major methodological critiques for the approach at this time. I do have a number of questions or comments relating to adding clarity to the approach, the findings, or their presentation, and I believe it is worthwhile that they be addressed before the manuscript is accepted for publication. It is my expectation that they can be relatively easily addressed, barring surprises I imagine the manuscript should be acceptable afterwards.

My comments and questions are as follows, generally in sequence with the manuscript.

We thank-you the reviewer for these comments and have addressed all the points below

- Abstract:

Results: The results note that both AMETHIST vs. KP program only and the KP program only vs discontinuation were cost-saving. It would be very informative to clarify when within the 50-year time horizon horizon cost savings were achieved.

We have added 'within 15 years' to the abstract.

- Introduction:

Line 114-115: Please share the magnitude and uncertainty (CI or p-value) of this risk reduction, as it is important context for the reader.

We have added the risk reduction of -5.5% to the Introduction, along with the CI

Line 117-118: Please share by what definition / threshold is being used when stating the KP program has been operating "cost-effectively"

Here, we are using the definition as referenced at the end of the sentence, i.e. based on previous analyses from the group, programs costing less than \$34 million are deemed to be cost effective. We have moved the reference to earlier on in the sentence to clarify this.

Methods:

255: The authors have (I assume inadvertently) placed the methods section at the end. It should be shifted to in between Intro and Results.

The Methods section was placed at the end to align with Nature Communications formatting guidelines

277 - 280: The logic of this cascade is reasonable. Further, it is appreciated that Table S27 is quite extensive and understood that within-manuscript space is always limited in available word counts to truly detail a full model. That said, changes to FSW-related parameters are the key experimental inputs for this investigation of predicted KP program effects, so I feel these disadvantage parameters deserve quantification and sourcing / justification within the main text. I believe either narrative or tabular form would be acceptable. If tabular, something akin to table A1 would be appropriate.

We are happy to move Table A1 to the main paper if editors agree, noting this table includes justification of changes made to parameter values.

283: Please spell out the acronym PHIA in its first usage

This has been done

285-287: Looking at A0, it looks like calibration was based on general population parameters. Were any FSW-specific indicators used to assess calibration? Overall, the calibration looks reasonable, but if no FSW-specific indicators were utilized, I would, either a) ideally check and report in A0 that they are also well calibrated, or b) discuss (here) as a limitation of the calibration that they either weren't available or did not calibrate as well as the general indicators in A0. [I believe this does get briefly mentioned later with respect to incidence, but further and earlier clarity would be appropriate.]

FSW specific outputs were also compared to observed data. The comparison is shown in Table 1 in the main text. We have added a note to Table A0 mentioning this.

290 - 296: Consider placing this paragraph in the Introduction rather than Methods section.

We included this paragraph in the Methods section to facilitate a clearer comparison between the observed KP programme and the modelled KP intervention. However, we are happy to move it to the Introduction if the editors would prefer.

303 - 305: I assume this refers specifically to the +AMETHIST; scenario. If so, please make that explicit.

The text in lines 303-305 is in reference to the KP programme without AMETHIST, i.e. a lower impact FSW program which does still reduce the disadvantages FSW may have, but at a lower magnitude than the + AMETHIST option.

309: Similar to my comment above, I'd put A1 or a version thereof into the main text near here. Adding / removing those specific parameters to the scenario runs is the experiment under investigation, and thereby warrant main text presentation.

We are happy to include Table A1 in the main text as suggested

Regarding A1, wherever placed, the reference to Cowan 2024 for the third column is much appreciated. To the extent that the second column (regarding KP program related parameters) relies on prior literature vs. assumption / broad sampling, any relevant citations should also be clearly noted.

We were able to reference the AMETHIST trial in the third column as the trial results clearly showed the impact of the trial for the specific outputs mentioned. For the KP parameters in the second column, parameters were sampled from distributions as shown to try and calibrate to data included in Table 1 in the main text. Hence we are unable to reference the parameters directly, but can, and have in Table 1 provided references to compare the outputs obtained as a result of parameter selection.

328 - 329: This manuscript would be strengthened by also providing the reported confidence interval around or a p-value for this difference.

We have added in the CI as requested

352: Please clarify that it is "projected" total costs that are compared

We have added in the word 'projected' in line 352.

353-359: This phrasing could be cleaned up. Nested parentheses are fine in math but a bad sign in writing, but more importantly it describes a net benefits calculation (in DALYs) but confusingly refers to it as calculating incremental DALYs informing a cost-effectiveness calculation, when in fact a cost-effectiveness ICER is never actually calculated.

The phrasing has been changed to:

The incremental DALYs averted by AMETHIST compared to the standard of care KP programme were calculated assuming a cost-effectiveness threshold of US\$ 500 per DALY averted. This was calculated as DALYs averted by AMETHIST less incremental costs divided by the cost effectiveness threshold.

Results:

162 - 163 & 173: The authors note that continuing the KP program is cost-saving compared to discontinuation within 20 years, and adding AMETHIST saves both costs and DALYs within 15 years. I think it is very worthwhile to specify specific years in which those achievements are realized. Both present-bias and genuine credit constraints can inform whether long-term positive returns will or even can be pursued by various decision-makers. With respect to USAID in particular, US assistance has for the short-run (and possibly longer) theoretically lended to treatment and at abdicated prevention. If available, reporting what year treatment costs alone under discontinuation surpass the combined treatment and prevention costs of continuing the KP program would be the most politically pertinent information to present to stakeholders.

These analyses were performed using 5 year increments and hence whilst we cannot specify the year exactly, we can say that in the AMETHIST vs. KP comparison, AMETHIST was cost-saving by 2040 and similarly in the discontinuation vs. KP comparison, the KP programme was cost-saving by 2040. We have added this to the Results.

Treatment costs alone under the discontinuation scenario are higher than treatment costs plus prevention costs in the continuation of the KP program scenario for much of the time horizon considered. This is most likely due to PrEP (now including long acting injectable forms) also being available outside of the FSW programme. The AMETHIST trial showed little impact on PrEP use in the AMETHIST arm, informing our modelling. We did see with our modelled outputs that discontinuation of the program resulted in the proportion of FSW with suppressed viral load dropping considerably and this would have impacted onward transmission of HIV significantly.

- DISCUSSION

200: Please re-phrase "would" eliminate issues to "could" eliminate issues.

This has been done

200-208: While it is indeed important to report that this analysis has not accounted for recently proven long-acting injectable PrEP formulations, the subsequent rush through related research by related authors and extrapolating projections therefrom

strays somewhat afield from the evidence generated in this investigation. I would summarize into a more brief statement along the lines that that cost effectiveness research on injectables implies that it is worth assessing under what circumstances and timelines they too might be cost-saving.

We have rerun the analyses now accounting for use of injectable long acting PrEP, assuming introduction of this in 2027. We have hence removed the limitations around PrEP from the Discussion.

226-235: Limitations of the primary AMETHIST analysis itself should also be acknowledged within these limitations, particularly that they are based on endline group comparisons absent baseline balance confirmation.

We have included a sentence on AMETHIST limitations reading as follows:

The trial also had notable limitations, including the exclusion of baseline data from the analyses and the reliance solely on endline group comparisons for outcome assessment (7).

Table 1: The phrasing "Percentage of SW with total duration in each category (years)" is unintuitive. Perhaps say, "Percentage of SW with total years spent as SW"?

This has been edited as suggested

Table 2: Does "Portion of SW tested in past year" refer to all SW, all HIV- SW, or all SW considered "At risk"?

We have clarified this throughout the table, either adding 'all', HIV negative' or 'HIV positive'

Also "Proportion of SW diagnosed" should be clarified to "Proportion of SW living with HIV diagnosed"

This has been changed as suggested

Table 3: I don't see lines referring to Non_ART programme costs, but they seem included to achieve \$149M & \$147M, correct? I recommend making at least one line for those.

Yes, all costs are included in the total costs. We had highlighted just the key costs (ART, testing) in the table but have now included a line for all other costs for clarity.

Supplementary Material:

I'd request that the Table of Contents for the Model Details section of the Appendix (p 5-6) further enumerate major headings within Table S27: Distribution of Parameters. For now there are 20+ pages of parameters to scan in search of the ones a reader

wishes to delve more into.

Further lines with key parameter headings have been added to the Contents as suggested

Table S29: Multiple of these unit costs come from 2014 papers. Please discuss if / how they were updated to reflect 2024 unit costs.

Broadly, costs have remained similar to those referenced, though unit costs do vary considerably according to source. We have added in a more recent reference to the costs in response to this comment: *Emily P Hyle, Thulani Maphosa, Ajay Rangaraj, Mary Feser, Geoffrey C Singini Clinical impact and cost-effectiveness of the WHO-recommended advanced HIV disease package of care. Lancet Glob Health. 2025 Aug;13(8):e1436-e1447*

Second response to reviewers

We appreciate the reviewers' careful consideration of our responses and their subsequent feedback.

I am grateful to the reviewers for their excellent rebuttal and for sharing the model documentation on the dedicated webpage. This resource will be valuable for the community interested in exploring the model in greater depth.

Thank-you – we agree the paper is much improved as a result of doing this.

I do, however, have a remaining concern regarding the decision not to fix the random seed. The authors responded:

“As such, each research question involves hundreds of parameters selected at random for each run. The model for this analysis was run 2000 times, and outputs across the HIV epidemic were considered suitable for inclusion in 100 of these runs, i.e. the calibration process. By setting a seed, we would be hugely restricting the uncertainty that we would like to convey across epidemic outputs. Instead, to reproduce the results, we would suggest running the model 2000 times as was originally done for these analyses, and analysing the approximately 100 runs that would pass the fit criteria coded in the core file and refined in the ‘create wide file’. We feel this would be much more useful than producing any one run using a set seed.”

I would like to clarify that my request was not about removing stochasticity from the model, which I fully recognize as essential in individual-based models. Rather, my concern relates to the reproducibility of the specific results presented in the manuscript. Running the model on a different computer or environment (e.g., a different SAS installation) could yield non-identical outputs, as random number generation may depend on local factors such as system time.

To address this, one possible approach would be to fix the initial random seed (from which the first number of a sequence is generated) to ensure reproducibility of reported results. Alternatively, comparable techniques could be applied to achieve the same goal. This would not constrain the stochasticity of the model overall, but it would allow independent researchers to replicate the precise outputs underlying the figures and analyses in the manuscript.

Reviewer #1 (Remarks on code availability):

Documentation and commenting of the model code have been much improved.

We appreciate the reviewers' concerns regarding reproducibility, as well as the

interesting suggestion to set an initial random seed. However, we believe that researchers aiming to reproduce our results would benefit more from replicating the entire modelling process including the incorporation of random effects as we have done. This approach offers a more comprehensive understanding of the methodology around individual based models and ensures alignment with our procedures. To support reproducibility and transparency, we have provided 90% uncertainty ranges for all model outputs, capturing both stochastic and parameter uncertainty. We believe that rerunning the model without fixing the random seed enables independent researchers to verify the robustness of our results by checking to see if their outputs align with ours.

Reviewer #2 (Remarks to the Author):

In this revised manuscript, the authors have done well at mostly addressing my previous comments and concerns. I have one remaining key request and a few minor edits.

The key request, as per my previous comments, is within-text presentation of the key parameters regarding FSW and how they are changed under different scenarios. As the parameters which vary between scenarios, they constitute the experiment being investigated and their presentation is important to conveying to the reader the nature of the investigation. These can be tabular and / or narrative, but must be within the main text. This information will come primarily from Table A1, but also the "parameters related to sex worker disadvantages" and "Parameters related to sex worker programs" sections of S27 on p 115 & 116 of the supplemental appendix. NOTE that A1 and s27, whilst concordant on most parameters, have *different values* for the effect of the KP program on adherence disadvantage reductions, being .1, .15, .25 in A1 but .2, .35, .5 in S27. These must be aligned with each other and the code as implemented. Relatedly, my understanding of the parameters as implemented is that there is discordance with their numeric presentations in A1 and S27 and the claim that on lines 307-309 that disadvantages were reduced, "negated, and/or resulted in an advantage" given that the multiplicative proportions in A1 and S27, applied to any disadvantage, cannot fully reach zero or change sign. These apparent discrepancies partially undergird my insistence that these parameters receive clear in-text attention.

We are happy to move Table A1 from the Appendix to the main text as suggested. Thank-you for spotting the discrepancy in Table S27 in relation to adherence disadvantages, this has now been corrected in Table S27 and aligns with the values presented in Table A1. With regards to the final point on discordance between the values in table A1 and the text on disadvantages '*being reduced, negated and/or resulted in an advantage*', this is correct; for example a FSW may have randomly

been assigned a base rate of interruption of 0.05/3 month period. Due to being a SW and having disadvantages, this rate may have increased say 2 fold (sampled) to 0.10. If she attended a program, the disadvantage would have a multiplicative factor applied to it, say 0.3 (sampled). This would result in the new interruption rate being $0.10 \times 0.3 = 0.03$, which is lower than the original base rate and would be considered as an advantage for the SW. We are happy to include this as a footnote to the table if Editors feel this would be helpful.

Otherwise, I feel my concerns have been addressed, with the following few and minor changes requested to finalize their editing.

1) I'd previously asked by what definition / threshold the authors were using the term "cost-effectively" on line 117. The authors have clarified that their definition was that utilized in reference 8. Looking at reference 8, I see the standard there is \$500/DALY averted. I request that the authors incorporate this benchmark (\$500/DALY) within the sentence describing the KP program as cost effective.

We have added a sentence in the Results as requested. Given that costs were saved and DALYs averted, this sentence may not be necessary but we are happy for the Editors to decide on this.

2) Line 285 points readers to A0 for evidence that modeled data calibrated well to empirical data. I see now that readers should be pointed to both A0 AND Table 1. The additional reference to further details in a differing publication is also appreciated, though the reference needs formatting.

We have added 'and Table 1' as suggested. The reference has also now been formatted.

3) I appreciate the authors' updating of the phrasing of lines 357-360. It is improved but still a bit incongruent with standard economic terminology by conflating net and incremental benefits. I would phrase this part as "The net benefits, in terms of DALYs averted, by AMETHIST compared to the standard of care KP program were calculated as the incremental DALYs averted by AMETHIST over the standard of care KP programme minus the incremental costs of AMETHIST divided by the assumed cost-effectiveness threshold of US\$500 per DALY averted."

We have reworded as suggested